# UNIT: Unifying Image and Text Recognition in One Vision Encoder

**Yi Zhu**[1], **Yanpeng Zhou**[1], **Chunwei Wang**[1], **Yang Cao**[2], **Jianhua Han**[1], **Lu Hou**[1], **Hang Xu**[1]

[1]Huawei Noah's Ark Lab, [2]Hong Kong University of Science and Technology

`https://github.com/yeezhu/UNIT.`

## Abstract

Currently, vision encoder models like Vision Transformers (ViTs) typically excel at image recognition tasks but cannot simultaneously support text recognition like human visual recognition. To address this limitation, we propose **UNIT**, a novel training framework aimed at **UN**ifying **I**mage and **T**ext recognition within a single model. Starting with a vision encoder pre-trained with image recognition tasks, UNIT introduces a lightweight language decoder for predicting text outputs and a lightweight vision decoder to prevent catastrophic forgetting of the original image encoding capabilities. The training process comprises two stages: *intra-scale pretraining* and *inter-scale finetuning*. During intra-scale pretraining, UNIT learns unified representations from multi-scale inputs, where images and documents are at their commonly used resolution, to enable fundamental recognition capability. In the inter-scale finetuning stage, the model introduces scale-exchanged data, featuring images and documents at resolutions different from the most commonly used ones, to enhance its scale robustness. Notably, UNIT retains the original vision encoder architecture, making it **cost-free** in terms of inference and deployment. Experiments across multiple benchmarks confirm that our method significantly outperforms existing methods on document-related tasks (e.g., OCR and DocQA) while maintaining the performances on natural images, demonstrating its ability to substantially enhance text recognition without compromising its core image recognition capabilities.

## 1 Introduction

Vision encoder models [6, 43, 44, 53, 41, 24] are crucial for extracting high-level visual features, thereby enhancing performance across a range of downstream tasks [9, 38, 8, 14, 57]. They play a vital role in integrating visual information into intelligent applications, driving advancements in both computer vision and artificial intelligence. In recent years, Vision Transformers (ViTs)based encoder models [16, 66, 27, 58, 71] have revolutionized the field of computer vision. ViT models pretrained on image classification tasks are extensively used as backbones for a wide array of image recognition tasks and achieves state-of-the-art performances. Another line of remarkable ViT models are trained via image-text cross-modal contrastive learning [43, 64, 52, 67]. These models are often used as plug-in vision encoders in Large-scale Vision-Language Models (LVLMs), which have emerged as versatile tools with the potential to revolutionize various domains, including healthcare diagnostics, autonomous driving, digital assistants, and advanced content analysis in media and entertainment. Despite these advancements, existing vision encoder models, exhibit a significant limitation: they have not yet demonstrated *the capability to simultaneously support both image and text recognition* like humans do. Such ability is essential for document analysis applications, for instance, a model must accurately recognize and interpret textual information embedded within complex layouts, such as tables, graphs, and mixed media.

38th Conference on Neural Information Processing Systems (NeurIPS 2024).

Text recognition [50, 49, 12], particularly in the context of dense documents and complex visual environments, presents unique challenges that are distinct from those encountered in pure image recognition. While image recognition typically focuses on global feature extraction and classification, text recognition requires precise local feature extraction and sequence prediction. ViTs, though adept at handling global image features, often struggle with the detailed local features needed for accurate text recognition, particularly in high-resolution document images where fine details are crucial. A straightforward solution is to finetune pre-trained ViTs using high-resolution documents. However, this approach requires interpolating the pre-trained positional embeddings to handle longer sequences. Such interpolation with a large magnification will disrupt the alignment between the original positional embeddings and their corresponding spatial positions, thereby impacting the model's performance.

Existing methods [39, 4, 5] build document-specific models by retraining ViTs exclusively on the Optical Character Recognition (OCR) task, thereby discarding the original image encoding capability. In downstream applications, these models are often ensembled with other expert models, requiring users to specify the data type in advance, which is inadequate for scenarios requiring dynamic and simultaneous recognition of both image and text without prior knowledge of the input content. Other ensemble-based methods [53, 54] use similar model architectures trained independently on image and text recognition tasks. Inputs are fed into both models, and the output features are concatenated to form a unified representation. However, this approach will result in a significant increase in computational cost. Additionally, some LVLM methods [62, 63] enhance document analysis tasks (e.g., DocQA) in an OCR-free manner by finetuning with document-instruction data. Nevertheless, their capabilities are limited to handling images with prominent text and fail with dense documents, struggling to generalize across different font sizes, typefaces, and backgrounds.

In this paper, we propose **UNIT**, a novel training framework for **UN**ifying **I**mage and **T**ext recognition abilities within a single model. UNIT upgrades an existing Vision Transformer (ViT) model to effectively integrate text recognition. First, a lightweight language decoder (e.g., OPT-125M) is introduced to decode the learned visual features into text sequences in an auto-regressive manner, enabling the encoder model to capture fine-detailed shape and sequential information necessary for text recognition. Then, a tiny vision decoder (e.g., two MLP layers) is introduced to reconstruct the visual features of the original vision encoder from the newly learned features, preventing catastrophic forgetting of the model's fundamental encoding ability for natural images. The training pipeline involves two stages. The first is the **intra-scale pretraining** stage, where the model takes images and documents at their commonly used resolutions as inputs, specifically *low-resolution images and high-resolution documents* (×4 times the low-resolution). During this stage, the model is optimized with three objectives: OCR for document inputs, feature reconstruction of the original ViT encoder's features and image captioning for natural image inputs. The second stage is **inter-scale finetuning** stage, where the model is trained under a scale-exchanged setting with *high-resolution images and low-resolution documents*, enhancing its scale robustness for both image and text recognition. It is important to note that UNIT retains the original vision encoder architecture, ensuring that there is no increase in inference cost.

Extensive experiments show that UNIT significantly outperforms document-specific models on OCR tasks while maintaining core image recognition capabilities. When integrated into LVLMs, it also benefits downstream document analysis tasks without degrading image understanding. This demonstrates that UNIT effectively unifies image and text recognition abilities.

The main contributions of this paper are summarized as follows:

- We propose UNIT, a novel framework that unifies image and text recognition in a single vision encoder through joint training of multi-tasks. The model retains the original vision encoder architecture, ensuring cost-free deployment and no increase in inference cost.

- The training paradigm comprises two key stages: an intra-scale pretraining stage, where images and documents are trained at their commonly used resolutions, and an inter-scale finetuning stage, where their resolutions are exchanged to enhance scale robustness.

- Extensive experiments demonstrate that UNIT significantly enhances performance on vision tasks requiring text recognition compared to their counterparts, while preserving the fundamental encoding ability on natural images.

## 2 Related Work

**Vision Transformer for Text Recognition.** In recent work, the application of Vision Transformers (ViT) to document reading has gained traction due to their success in image recognition [30, 10, 24, 4, 15]. Nougat [5]leverages a ViT model for Optical Character Recognition (OCR) tasks, focusing on converting human-readable scientific documents into a machine-readable markup language. Similarly, KOSMOS-2.5 [19] employs a ViT-based vision encoder coupled with a Transformer-based language decoder, aiming to serve as a versatile tool for diverse text-intensive image understanding tasks through supervised fine-tuning. The Donut model [23], while introducing an OCR-free transformer for visual document understanding, may encounter challenges in generalizing to unfamiliar document types. Despite their demonstrated efficacy in specialized OCR scenarios, these models may not fully harness the original image encoding capabilities present in pre-trained ViT architectures. This limitation can restrict their applicability to text-only images.

**LVLMs for Document Analysis.** Several recent studies have explored OCR-free visual-situated language understanding using Large Vision-Language Models (LVLMs) [47, 55, 59, 1, 2, 70, 40, 28], where document-instruction data is used to fine-tune LLMs without adapting the vision encoder [62, 60, 61]. These methods often rely on a pretrained Vision Transformer (ViT) as a fixed vision encoder, which may present several challenges: 1) The frozen ViT may limit text recognition capabilities, especially when documents feature varied fonts, styles, or low image quality not seen during its pretraining. 2) The model's reliance on a constant image resolution can lead to inaccuracies in text recognition for high-resolution documents or those with noise and distortion. Additionally, several recent works [53, 54] have successfully enhanced LVLMs with OCR-like abilities, by retraining a document-specific Vision Transformer (ViT) and then concatenating its output visual features with those from a pretrained ViT model. However, this method comes with trade-offs, notably an increase in computational expense and a significant underutilization of the model's capacity, leading to inefficiencies. In contrast, our UNIT integrates both capabilities within a single model, offering greater efficiency, cost-free deployment, and no increase in inference time.

**Multi-scale Vision Transformer.** Vision Transformer models are commonly pre-trained to process images at a fixed resolution, such as 224 or 336 pixels, which is commonly used for many image understanding tasks [17, 20, 33, 26, 25]. However, these resolutions might be insufficient for precisely discerning tightly packed text in higher-resolution documents. Consequently, a multi-scale training approach for Vision Transformers is crucial for effectively managing both image and text recognition challenges. The progression of Vision Transformer architectures has embraced a notable trend toward multi-scale modeling. This paradigm enables the models to capture and process visual information across different scales, enhancing their capability to understand complex scenes and texts. Some methods [29, 51, 56] integrate a hierarchical pyramid structure into the original Vision Transformer architectures, which continue to operate at a constant resolution. In contrast to these methods, our proposed UNIT model breaks the limitations of fixed-resolution inputs. UNIT is designed to handle multi-scale training for multi-tasks, yielding a unified representation with scale robustness for both images and documents.

## 3 Methodology

### 3.1 Preliminaries

**Backbone.** We employ the Vision Transformer (ViT) [16] architecture as our backbone. Let $f_\theta(\cdot)$ be the vision encoder with parameters $\theta$, for an input image $\mathbf{I} \in \mathbb{R}^{H \times W \times 3}$, where $H$ and $W$ represent the image height and width, respectively. ViT model first partitions each image into fixed-size of patches in $p \times p$ pixels, and then encode these patches into hidden features of dimension $d$. Subsequently, the features are added with their corresponding position embeddings and fed into multiple layers of stacked Transformer blocks, interacting with each other via attention mechanisms. Finally the model outputs visual tokens $\mathbf{X} \in \mathbb{R}^{N \times d}$, as:

$$\mathbf{X} = f_\theta(\mathbf{I}) = \{\mathbf{x}_i\}_{i=1}^N, \tag{1}$$

where $N = N_s + N_c$ denotes the total number of visual tokens, comprising the spatial tokens $N_s = \frac{H}{p} \times \frac{W}{p}$ and the number of [CLS] tokens $N_c$.

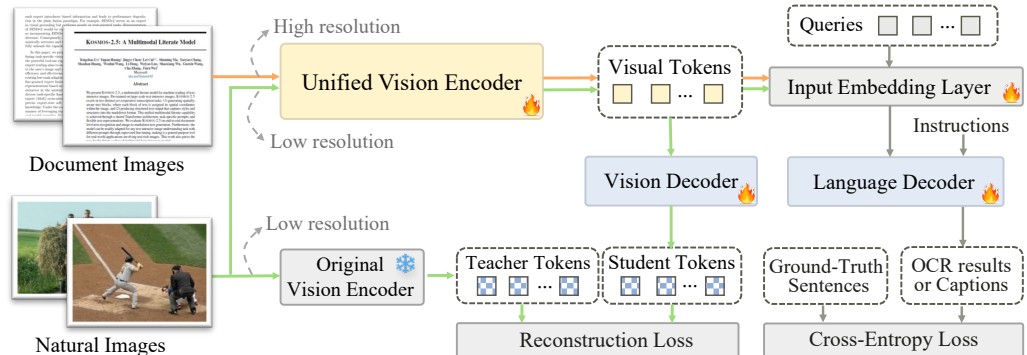

Figure 1: Overview of UNIT Architecture. The model processes high-resolution documents and low-resolution images, generating a set of visual tokens. These tokens pass through an input embedding layer, with document tokens fed into the language decoder to predict text sequences, enhancing the model's text recognition capability. Simultaneously, to preserve the model's original image encoding ability, the visual tokens from natural images are reconstructed via a lightweight vision decoder, mimicking the output of the teacher model. Additionally, an image captioning task is included alongside the OCR task to further enhance image understanding.

**Architecture.** UNIT aims to unify image and text recognition capabilities within a single ViT model, as illustrated in Fig. 1. This is achieved through the introduction of a lightweight language decoder (e.g., OPT-125M) for document-level OCR, enabling the ViT model to recognize text. Additionally, to preserve the original image recognition capabilities of the vision encoder, UNIT incorporates a vision decoder responsible for token-wise feature reconstruction from the original ViT model. To further enhance image understanding, an image captioning task is integrated alongside the OCR task. UNIT is trained in a multi-scale setting, where images and texts are processed at their commonly used resolutions during pretraining and then finetuned at other resolutions.

## 3.2 Text Recognition Ability Enhancement

**Multi-Scale Inputs.** In contrast to image recognition that focuses on global feature extraction and classification, text recognition requires fine local feature extraction and sequence prediction. ViTs are excellent at global feature extraction but may not perform well in text recognition. The reason is that ViTs use a learned position embedding for each input patch in an image, which in turn forces that the model always operate at a constant resolution, typically chosen as 224, 256 or 336. At these resolutions, the generated visual features may lose critical information, leading to difficulties in recognizing characters from dense texts such as PDF documents, websites, or digital books. Naively cropping high-resolution document images into low-resolution inputs would break the sequential order of characters and may impair the recognition of letters at the cutting boundaries. To prevent these issues, we introduce the Cropped Position Embedding (CPE) [22] augmentation, which interpolates the original position embeddings to match the number of positions of the maximum input size. For low-resolution images, the position embeddings are then randomly cropped and interpolated to match the number of input patches for the original model.

**Language Decoder.** Conventional methods often employ CLIP-style contrastive learning to align images with their language annotations. However, this approach presents two notable drawbacks in our context. Firstly, contrastive learning relies on a CLIP pretrained text encoder to generate language embeddings. The maximum sequence length of 77 tokens is insufficient for text recognition annotations, especially in dense documents. Secondly, such training paradigm is inadequate for the vision encoder to accurately capture each word, leading to undesirable information loss during the encoding process. Taking the above factors into account, we introduce a lightweight Transformer decoder, e.g., OPT-125M [69], to efficiently predict the language sequences presented in the input documents. Similar to LVLMs, we employ an input embedding layer to project the visual features from the vision encoder into the embedding space of the language decoder. Here we initialize the input embedding layer with a two-layer Q-Former [25] with $K$ learnable query tokens $\mathbf{Q} = \{\mathbf{q}_k\}_{k=1}^K, \mathbf{q}_k \in \mathbb{R}^d$. The visual tokens $\mathbf{X} = \{\mathbf{x}_i\}_{i=1}^N$ generated from the vision encoder are first fed into the input embedding layer and then the language decoder. We denote this process as $f_\phi(\cdot)$ and $\phi$

denotes the parameters of the input embedding layer and the language decoder:

$$\{\mathbf{z}_1, ..., \mathbf{z}_N\} = f_\phi(\{\mathbf{x}_1, ..., \mathbf{x}_N\}, \{\mathbf{q}_1, ..., \mathbf{q}_K\}). \tag{2}$$

The decoder uses the last hidden state $\mathbf{z}_t^L \in \mathbb{R}^d$ at time step $t$ to predict language sequence in the form of text tokens $\hat{\mathbf{y}} = \{\hat{y}_1, \hat{y}_2, ..., \hat{y}_T\}$ via the language decoder in an auto-regressive manner. The cross-entropy loss for training the auto-regressive Transformer decoder is defined as follows:

$$\mathcal{L}_{\text{lan}}(\mathbf{y}, \hat{\mathbf{y}}) = -\sum_{t=1}^{T} \log P(\hat{y}_t = y_t | \mathbf{y}_{1:t-1}, \mathbf{z}_t^L), \tag{3}$$

where $\mathbf{y} = \{y_1, y_2, ..., y_T\}$ represents the target sequence, and the probability $P(\hat{y}_t = y_t | \mathbf{y}_{1:t-1}, \mathbf{z}_t^L)$ is obtained from the softmax output of the final layer of the decoder.

### 3.3 Image Recognition Ability Preservation

**Vision Decoder.** Finetuning the vision encoder exclusively on document datasets would result in severe catastrophic forgetting of its original image encoding capabilities. To mitigate this, we introduce an token-wise feature reconstruction task on natural image datasets, ensuring that the newly learned vision encoder retains its ability to encode visual concepts. We incorporate a visual decoder $f_\pi(\cdot)$ consisting of two fully connected layers with a GeLU activation function in the intermediate layer. The decoder processes each visual token independently, preserving the positional information inherent in each token more effectively. We utilize the original pretrained ViT model as a teacher model to provide original features for each natural image, providing supervision signals for visual feature reconstruction. Denoting the teacher tokens as $\hat{\mathbf{X}} = \{\hat{\mathbf{x}}_i\}_{i=1}^N, \hat{\mathbf{x}}_i \in \mathbb{R}^d$, we enforce the alignment of the new learned student tokens with the original ones using a weighted sum of the cosine distance $L_{cos}$ and smooth L1 loss $L_{l1}$. This approach ensures consistency in both vector direction and magnitude between the new and original features, as:

$$\mathcal{L}_{\text{vis}}(\mathbf{X}, \hat{\mathbf{X}}) = \sum_{i \in \mathcal{C}} L_{\cos}(f_\pi(\mathbf{x}_i), \hat{\mathbf{x}}_i) + \mu L_{l1}(f_\pi(\mathbf{x}_i), \hat{\mathbf{x}}_i), \tag{4}$$

where $\mathcal{C}$ represents a subset of features randomly sampled from the $\hat{\mathbf{X}}$, to mitigate overfitting and enhance robustness during training. $\mu$ denotes the loss weight scalar.

**Language Decoder.** Furthermore, we incorporate an image captioning task alongside the text recognition task to ensure that the learned visual tokens can be effectively projected into the language space, thereby enhancing image understanding ability. The forward process and loss formulation for image inputs are the same as those for document inputs.

### 3.4 Intra-Scale Pretraining

As described in Sec. 3.2 and Sec. 3.3, UNIT tackles images and documents at different input scales with different optimization objectives. In this section, we introduce the intra-scale pretraining stage (see Fig. 2a) where image and text recognition are trained at a fixed scale, namely, low resolution for images and high resolution for documents, without considering variations in scale.

**Dataset Preparation.** Let $\mathcal{D}$ be a curated dataset with 5M samples, where $\mathcal{D} = \mathcal{D}_{\times 1}^I \cup \mathcal{D}_{\times 4}^T$. Here, $\mathcal{D}_{\times 1}^I$ represents a dataset of natural images annotated with coarse captions (less than 30 words). The images are resized to $\times 1$ times the original scale $r$, primarily sourced from the Conceptual Caption dataset [46], comprising 3M samples. Meanwhile, $\mathcal{D}_{\times 4}^T$ represents a dataset of documents annotated with dense OCR data (more than 500 words). The document images are resized $\times 4$ times the original scale $r$, sourced from our synthetic dataset of English PDF documents, comprising 2M samples.

**Instruction Prompts.** The instruction prompts follow the format used in popular LVLMs [70, 25, 1, 3, 2], where image tokens are prefixed with text tokens. Specifically, we use two special tokens "" and "</img>", to indicate the start and the end position of the image tokens, followed by instructions that indicate task requirements. For natural images, the prompt is set as "Give a caption of this image:" and the LLM outputs a language sequence summarizing the content of the image. For document images, the prompt is set as "Read the text in this image:" and the LLM outputs sentences row by row as they appear in the document in the order of reading.

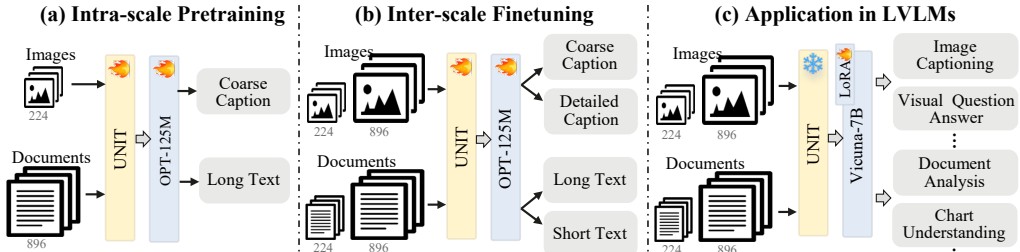

Figure 2: Illustration of the UNIT training paradigm. The (a) intra-scale pretraining stage processes images and documents at their commonly used resolutions to integrate basic text recognition with existing image recognition capabilities. The (b) inter-scale finetuning stage processes scale-exchanged data and tasks to enhance scale robustness, benefiting downstream document analysis tasks when integrated into (c) LVLMs applications.

**Joint Optimization.** We train all parameters, including the ViT backbone, vision decoder and language decoder, using images and documents annotated with language sentences. During training, the model is optimized by the cross-entropy loss $\mathcal{L}_{\text{lan}}$ between language outputs and annotations, and also the feature reconstruction loss $\mathcal{L}_{\text{vis}}$ to force the newly learned visual features to be similar with the original ones. The UNIT model is updated to minimize the total loss:

$$\theta = \arg\min_{\theta} \mathcal{L}_{\text{lan}}(\{\mathcal{D}^I_{\times 1} \cup \mathcal{D}^T_{\times 4}\}) + \lambda \mathcal{L}_{\text{vis}}(\{\mathcal{D}^I_{\times 1}\}). \tag{5}$$

## 3.5 Inter-Scale Finetuning

After the intra-scale pretraining stage, UNIT is capable of processing both image and text recognition at their commonly used resolutions, namely, low-resolution images and high-resolution documents. However, we observed that this model lacks scale robustness when handling texts with larger fonts and images with larger dimensions, significantly limiting its generalization ability in various vision tasks. Naively add the scale-exchanged datasets in the intra-scale pretraining process may slow down the convergence of the model and making it prone to local optima. To address this, we further conduct inter-scale finetuning (see Fig. 2b), using high-resolution images and low-resolution documents.

**Dataset Preparation.** Here we build another scale exchanged dataset $\mathcal{D}^* = \mathcal{D}^I_{\times 4} \cup \mathcal{D}^T_{\times 1}$ with 1M samples for the inter-scale finetuning. We conduct a dataset $\mathcal{D}^I_{\times 4}$ comprising natural images annotated with detailed captions (each containing over 100 words). These images are resized to $\times 4$ times the original scale $r$, primarily drawn from the ShareGPT4V dataset [11], totaling 1M samples. Additionally, we create a dataset $\mathcal{D}^T_{\times 1}$ comprising 1M documents with an average font size $\times 4$ times larger than that of $\mathcal{D}^T_{\times 4}$. The document images are resized $\times r$ times, sourced from our synthetic dataset which includes cropped pages from digital books, advertisements, websites, and other sources containing short yet large font texts. We adopt the instruction prompt setting same to the intra-scale pretraining stage.

**Random Feature Sampling.** When dealing with natural images at higher resolutions, the resulting increase in visual tokens compared to lower resolutions can lead to gradient overflow during the training process, primarily due to the token-wise feature reconstruction loss. To address this challenge, we propose to randomly sample a set of visual tokens equal to the number in the low-resolution inputs and discarding other tokens.

**Joint Optimization.** This stage aims to enhance the scale-robust recognition ability for both image and text. We retain half of the data at the previous stage and then introduce the scale-exchanged data. The model is updated to minimize the total loss:

$$\theta = \arg\min_{\theta} \mathcal{L}_{\text{lan}}(\{\mathcal{D}^I_{\times 4} \cup \mathcal{D}^T_{\times 1} \cup \frac{1}{2}\mathcal{D}^I_{\times 1} \cup \frac{1}{2}\mathcal{D}^T_{\times 4}\}) + \lambda \mathcal{L}_{\text{vis}}(\{\frac{1}{2}\mathcal{D}^I_{\times 1} \cup \mathcal{D}^I_{\times 4}\}). \tag{6}$$

# 4 Experiments

**Implementation Details.** We select OpenCLIP ViT-H [43] with 32 layers and a hidden size of 1280 as our vision encoder. We choose OPT-125M [68] model with a hidden size of 768 as the language

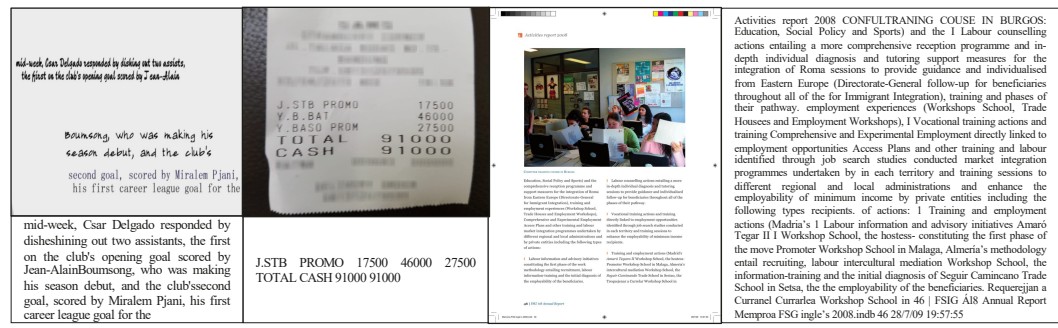

Figure 3: Visualization examples of text recognition. UNIT predicts accurate OCR results even across diverse scenarios, e.g., handwritten texts, receipts, and interleaved image-text documents. Please see clearly by zooming in. More promising examples are shown in the supplementary material.

| Method | Backbone | #Params | Input | FUNSD | SROIE | CORD | SYN-L-val | MD-val |
|---|---|---|---|---|---|---|---|---|
| Donut [23] | Swin-B | 260M | $1280 \times 960$ | 9.08 | 8.94 | 16.64 | 44.78 | 5.07 |
| Nougat [5] | SAM-ViT-B | 247M | $896 \times 672$ | 55.35 | 33.64 | 1.57 | 66.76 | **86.71** |
| Vary* [53] | SAM-ViT-B | 525M | $1024 \times 1024$ | 21.01 | 9.84 | 12.89 | 91.20 | 59.30 |
| RADIO* [44] | ViT-H/14 | 632M | $896 \times 896$ | 26.12 | 10.42 | 10.01 | 93.90 | 37.57 |
| UNIT (ours) | ViT-H/14 | 632M | $896 \times 896$ | **67.14** | **41.48** | **58.87** | **95.33** | 78.50 |

Table 1: Comparison with ViT-based models for text recognition ability. The presented numbers are F1 values. An asterisk (*) indicates reimplemented results on our document datasets.

decoder in the two training stages. Our optimization strategy involves AdamW [32] with a weight decay 0.01. The initial learning rate to 5e-5, and changes with a cosine learning rate decay scheduler. The warmup ratio is set to 0.03, and the global batch size is 256. We set loss weights $\lambda = 2$ and $\mu = 0.2$. These settings are shared for both two training stages. Our model can be trained with any resolution below the maximum limit. We chose two primary resolutions for training to ensure consistent tensor sizes, simplifying batch processing and optimizing resource use. UNIT is trained on these two resolutions and tested on different ones. To ensure efficient data loading and processing during training, we preprocessed the document images by padding them to square shapes. During inference, we maintain the original aspect ratio of the input images.

## 4.1 Evaluation Benchmarks

**Text Recognition: 1) Document-level OCR:** We evaluate our model on three public OCR datasets: FUNSD (50 images), SROIE (347 images), and CORD (100 images), reporting the F1 score. Given the lack of large blocks of dense text in these datasets, we create two additional OCR evaluation datasets using English PDF files from arXiv. The first, SYN-L-val (200 images), consists of PDFs with small font sizes at approximately 1k resolution. The second, SYN-S-val (200 images), includes images cropped from these PDFs, resized to a lower resolution (e.g., 224), resulting in larger font sizes. **2) Markdown conversion:** The task requires structured text output from images of documents, capturing the stylistic and structural elements of the text in a markdown format. We add 1M markdown data (collected following Nougat [5]) into the inter-scale finetuning stage, endowing the ability of markdown conversion of our model. Due to the lack of cleaned markdown conversion validation set, MD-val, we conduct a dataset containing 82 images with markdown format annotations.

**Image Recognition: 1) Zero-shot classification:** We calculate the feature similarity between the [CLS] token of the visual encoder and the text features extracted from the corresponding CLIP text encoder. The evaluation is performed on the ImageNet-1K dataset, and top-1 accuracy is reported. **2) k-NN classification:** We first compute the [CLS] token of the visual encoder for the training images of ImageNet-1K, and then for each validation image, we utilize a weighted sum of the $k$ nearest training vectors to select a label. **3) Semantic Segmentation:** we append a linear head onto the vision encoder and train it with the encoder frozen. The AdamW optimizer is used with a learning rate of $10^{-3}$. The segmentation mIoU (%) is computed as in the MMSeg framework [13] on the ADE20k dataset. For both training and evaluation, we use an input size of $512 \times 512$.

| Method | #Param. | ZS cls. | kNN cls. | Segm. |
|---|---|---|---|---|
| EfficientViT-L1 [7] | 38M | 71.73 | 79.90 | 33.12 |
| SwinV2-S [29] | 49M | 74.70 | 81.12 | 35.57 |
| ConvNext-B [31] | 88M | 75.43 | 81.73 | 38.95 |
| MViTV2-B [27] | 51M | 75.92 | 81.39 | 41.39 |
| NFNet-F3 [6] | 254M | 76.93 | 80.50 | 38.31 |
| MaxViT-B [48] | 119M | 77.49 | 79.34 | 38.46 |
| OpenCLIP-H/14 [43] | 632M | 77.19 | 81.10 | 40.04 |
| RADIO-L/14 [44] | 304M | 77.25 | 84.03 | 48.70 |
| E-RADIO-L/14 [44] | 265M | 77.87 | 83.73 | 45.50 |
| RADIO-H/14 [44] | 632M | 78.62 | 84.17 | 49.01 |
| UNIT (ours) | 632M | **78.76** | **84.18** | **50.19** |

Table 2: Comparison of image recognition capabilities with existing vision encoders.

| Method | ZS cls. | OCR SYN-L-val | | |
|---|---|---|---|---|
| | | Prec. | Rec. | F1 |
| *w/o Image captioning* | | | | |
| $\lambda = 0$ | N/A | 93.03 | 90.63 | 91.81 |
| $\lambda = 1$ | 74.20 | 92.18 | 89.43 | 90.78 |
| $\lambda = 2$ | 76.24 | 94.13 | 91.72 | 92.91 |
| *with Image captioning* | | | | |
| $\lambda = 0$ | N/A | 95.17 | 93.39 | 94.26 |
| $\lambda = 1$ | 75.16 | 92.56 | 89.26 | 90.88 |
| $\lambda = 2$ | **78.54** | **95.45** | **93.72** | **94.57** |

Table 3: Ablation of the feature reconstruction loss weight $\lambda$ with both the image captioning task during the intra-scale pretraining stage.

| Method | Commonly-used Resolution | | | | Exchanged Resolution | | | |
|---|---|---|---|---|---|---|---|---|
| | ZS cls. | OCR Prec | OCR Rec. | OCR F1 | ZS cls. | OCR Prec | OCR Rec. | OCR F1 |
| w/o inter-scale | 78.54 | 95.45 | 93.72 | 94.57 | 0.40 | 51.79 | 28.53 | 34.99 |
| w/o feat. sample | 78.02 | 93.96 | 86.63 | 90.14 | 31.66 | 96.23 | 89.93 | 92.40 |
| UNIT (ours) | **78.76** | **96.67** | **94.03** | **95.33** | **80.49** | **96.35** | **90.61** | **92.68** |

Table 4: Ablation of the inter-scale finetuning stage and the random feature sampling strategy.

**Downstream Vision Tasks:** We replace the vision encoder in the LLava-1.5 [28] setting with our own encoder. We first pretrain the input embedding layers with the vision encoder and the LLM frozen, then conduct instruction tuning to finetune a Vicuna-7B model [42]. The learning rate of stage 3 is set to 5e-4 and a batch size of 512. We evaluate the trained models on general visual question answering (VQA) datasets including VQAv2 [17], GQA [20], and OKVQA [33], and also document comprehension datasets including ChartQA [34], DocQA [36], and InfoVQA [35].

## 4.2 Comparison with Existing Approaches

**Text Recognition.** We compare UNIT with two ViT-based OCR expert model, including Donut [23] and Nougat [5], and two open-source vision encoders that support high-resolution inputs, including Vary [53] and RADIO [44]. We use the open-source weights of Donut-Base and Nougat for evaluation. As for Vary and RADIO, we retrain them on our dataset for a fair comparison. The models are evaluated on public OCR dataset like FUNSD [21], SROIE [19], CORD [19] and our synthetic datasets SYN-L-val and MD-val. As shown in Tab. 1, UNIT consistently outperforms these methods on OCR and markdown conversion tasks, except for Nougat, exhaustively trained on markdown data. This shows UNIT's superior ability to encode fine-detailed sequential and spatial information in documents, resulting in highly accurate OCR outcomes, see Fig. 3. Based on the fundamental text recognition ability, UNIT can be further finetuned with a small amount of data to adapt to highly specialized tasks, such as text recognition with grounding (in both natural images and pure texts), markdown conversion and Chinese OCR. Please refer to the supplementary for more details.

**Image Recognition.** To validate the image encoding ability of UNIT, we compare it with existing efficient vision encoders on zero-shot and kNN classification on ImageNet-1k, as well as a linear prob semantic segmentation benchmark. In Tab. 2, for fair comparison, all models share the same teacher knowledge, derived from DINOv2 and OpenCLIP-H. The compared models directly use these two models for training feature reconstruction, and UNIT employs one of the best-performing students, RADIO-H, as the teacher encoder. From the results shown in Tab. 2, we observe that UNIT achieves the best performance among all these models, demonstrating its ability to effectively preserve the fundamental image encoding capabilities.

## 4.3 Ablation Studies

**About intra-scale pretraining.** As shown in Table 3, removing the additional image captioning task described in Sec. 3.3 leads to a significant drop in performance on zero-shot classification (from 78.54% to 76.24%). This demonstrates that the image captioning task can further enhance image recognition capabilities. Additionally, we ablate the feature reconstruction loss weight $\lambda$, finding

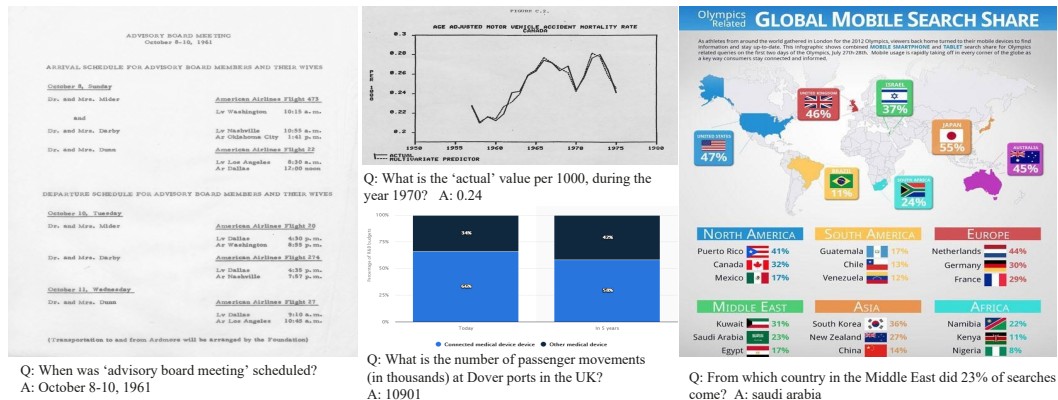

Q: When was 'advisory board meeting' scheduled?
A: October 8-10, 1961

Q: What is the 'actual' value per 1000, during the year 1970?  A: 0.24

Q: What is the number of passenger movements (in thousands) at Dover ports in the UK?
A: 10901

Q: From which country in the Middle East did 23% of searches come?  A: saudi arabia

Figure 4: Visualization examples of downstream document analysis tasks. UNIT accurately recognizes tiny words and digits, providing correct answers for document-related questions from users.

| Benchmark | $224 \times 224$ | $336 \times 336$ | $448 \times 448$ | $672 \times 672$ | $896 \times 896$ |
|---|---|---|---|---|---|
| Zero-shot classification | 78.8 | 81.3 | 81.7 | 81.3 | 80.5 |
| OCR F1 | 92.7 | 93.7 | 93.2 | 93.9 | 95.3 |

Table 5: Ablation studies on different input resolutions during inference.

the best performance when $\lambda = 2$. When the feature reconstruction loss $\lambda = 0$, the zero-shot classification failed. This is reasonable because zero-shot classification evaluates the alignment between the vision and text encoders. If $\lambda = 0$, the vision encoder cannot ensure proper alignment with the text encoder of OpenCLIP-H, leading to the failure of zero-shot classification.

**About inter-scale finetuning.** As shown in Table 4, "w/o inter-scale" represents the model trained without inter-scale finetuning. Commonly-used Resolution indicates that the input sizes are at the common resolution of the evaluated tasks, e.g., $224 \times 224$ for zero-shot classification and $896 \times 896$ for document OCR (SYN-L-val). Exchanged Resolution indicates $896 \times 896$ for zero-shot classification and $224 \times 224$ for document OCR (SYN-S-val). We can see that after inter-scale finetuning, UNIT demonstrates enhanced scale robustness on the exchanged resolution for both image and documents. "w/o feat. sample" represents the model trained without random feature sampling mentioned in Sec. 3.5. This setting causes a dramatic performance drop on zero-shot classification on at exchanged resolution, showing the effectiveness of random feature sampling.

**Different resolutions.** Our model supports arbitrary input sizes within the maximum limit. As shown in Table 5, our model, trained on two primary resolutions, generalizes well and maintains consistent performance in both zero-shot classification and OCR tasks.

### 4.4 Downstream Task Performance

**Naive resolution.** Vision encoders act as a key component in Large Vision-Language Models (LVLMs) to provide rich and detailed visual representations that enable effective cross-modal understanding and reasoning. Following LLava-1.5 [28], we build LVLMs using UNIT or other vision encoders based on the Vicuna-7B LLM. The vision encoders are fixed during LVLM training. Images are padded and resized to one of two primary resolutions based on the task requirements. A lower resolution (e.g., 224x224) is used for global understanding tasks (e.g., VQA), while a higher resolution (e.g., 896x896) is used for tasks requiring fine-detail perception (e.g., DocQA). Each image is represented as 256 visual tokens through the input embedding layer. The LLM is finetuned with LoRA [18]. The multi-modal pretraining process uses 4M image-caption data (randomly extracted from Conceptual Caption [46] and the LAION-COCO [45]) and 200k document OCR data (randomly sampled from the training set). In the Supervised Finetuning (SFT) stage, we use the LLaVA-80k or LLaVA-CC665k along with the train set of DocVQA [36] and ChartQA [34] as the fine-tuning dataset. In Tab. 6, UNIT significantly outperforms the teacher model RADIO and other compared models on the document analysis tasks, including ChartQA, DocQA and InfoVQA, while maintains comparable performance on image understanding tasks. From Fig. 4 we can see that UNIT can extract the texts in documents, charts or websites, showing its potential in real-world document analysis applications.

| Method | #Param | Document Analysis | | | Image Understanding | | |
|---|---|---|---|---|---|---|---|
| | | DocQA | ChartQA | InfoVQA | VQAv2 | GQA | OKVQA |
| CLIP-L [43] | 304M | 22.3 | 23.6 | 25.0 | 72.8 | 61.2 | 55.8 |
| DINOv2 [41] | 632M | 14.7 | 15.9 | - | - | 63.9 | - |
| SAM-L [24] | 632M | 13.9 | 15.0 | - | - | 51.0 | - |
| Vary [53] | 525M | 42.8 | 41.8 | - | - | 42.6 | - |
| RADIO-H [44] | 632M | 18.2 | 20.2 | 24.5 | 72.9 | 61.8 | 55.9 |
| **UNIT (ours)** | 632M | **43.0** | **46.0** | **28.7** | 72.3 | 60.4 | 55.5 |

Table 6: Comparison of the performance of various vision encoders with naive resolution.

| Method | ChartQA | DocVQA | InfoVQA | OCRBench | GQA | OKVQA | MME | MathVista |
|---|---|---|---|---|---|---|---|---|
| CLIP-L [43] | 52.0 | 57.2 | 29.3 | 382 | 62.3 | 57.0 | 1503.6 | 42.7 |
| SigLIP [65] | 56.5 | 62.0 | 29.7 | 429 | 63.0 | 61.1 | 1489.4 | 44.2 |
| **UNIT (ours)** | **61.0** | **65.5** | **31.9** | **480** | **63.9** | **61.5** | **1529.8** | **44.6** |

Table 7: Comparison of commonly used vision encoders in LVLMs with high-resolution grid slicing.

**High resolution.** In the naive resolution setting, each image is represented using 256 tokens. However, this may be insufficient for capturing the details of images with extensive content, such as densely organized documents containing thousands of words. To address this, we evaluate UNIT in Llava-Next with a grid slicing strategy and compare it with two commonly used vision encoders in LVLMs. This method divides each image into several grids based on their aspect ratio. For instance, with a maximum of 4 grids, possible configurations include $\{2 \times 2, 1 \times 2, 1 \times 3, 2 \times 1, 3 \times 1\}$, with each grid sized at $M \times M$. Here, $M$ corresponds to the resolution used during model pretraining—336 for CLIP-L and 384 for SigLIP. And we set 448 for UNIT. After the vision encoder, we use C-Abstractor [37] as the input embedding layer. It generates 256 visual tokens for each grid, which are then concatenated to create the final image representation. During the multi-modal pretraining stage, the input embedding layer is initially trained on Llava665k. Following this, the LLM and the last few layers of the vision encoder are trained using 1.5M image captioning data. In the SFT stage, all parameters of the LVLM are fine-tuned using 1.2M SFT data. As shown in Tab. 7, our method significantly outperforms the compared models on document-oriented QA tasks and demonstrates comparable performance on other QA tasks. CLIP-L and SigLIP are initially pretrained only on natural images and subsequently fine-tuned with document images for downstream tasks. In contrast, UNIT is pretrained on both natural images and documents using fundamental recognition tasks such as image captioning and OCR. Our approach offers two key advantages: First, pretraining with fundamental recognition tasks enhances UNIT's versatility and suitability for downstream document tasks. Second, the early integration of text recognition capabilities in UNIT provides a robust initialization for LVLM training, leading to more stable training and faster convergence.

## 5   Conclusion

We propose UNIT, a simple yet effective framework that unifies image and text recognition in a single vision encoder, without increasing deployment or inference cost. UNIT incorporates a lightweight language decoder for text recognition and a lightweight vision decoder to preserve image recognition capabilities. Trained with multi-scale inputs, UNIT processes images and documents at their conventional resolutions during intra-scale retraining for basic recognition, and at exchanged resolutions during inter-scale finetuning to enhance scale robustness. Extensive experiments demonstrate that UNIT significantly outperforms document-specific models, maintains core image encoding ability, and benefits downstream document analysis tasks when integrated into LVLMs, showing great potential in processing interleaved text and image information in real-world scenarios.

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

# A  Appendix / supplemental material

## A.1  Computer Resources

We implement our method using PyTorch 2.1.0 with CUDA 11.7. The training process requires 128 Ascend 910B GPUs (each with 64 GB of memory). Each training stage runs for one epoch, totaling 150 hours of training time.

## A.2  Limitations

Integrating text and image recognition in a single model may entail trade-offs, potentially yielding slightly inferior performance in highly specialized tasks compared to models exclusively dedicated to text or image recognition. For example, as illustrated in the Table 1 of our paper, UNIT's performance in the markdown conversion task (a highly specialized OCR task) falls short of Nougat, a model exhaustively trained on markdown data and specifically optimized for markdown conversion.

## A.3  Broader Impacts

**Positive Societal Impacts:** 1) Enhanced Accessibility: By unifying image and text recognition within a single model, UNIT could potentially improve accessibility for individuals with visual impairments. The model's ability to accurately recognize and interpret both images and text could facilitate the development of assistive technologies such as screen readers and optical character recognition (OCR) systems. 2) Efficiency in Document Analysis: UNIT's capability to seamlessly handle both image and text recognition tasks could streamline document analysis processes in various domains, including healthcare, education, and legal sectors. This could lead to increased efficiency and productivity in tasks such as document digitization, content extraction, and information retrieval. 3) Technological Advancement: The development of novel training frameworks like UNIT contributes to the advancement of computer vision and natural language processing technologies. Such advancements have the potential to drive innovation in diverse fields, including artificial intelligence, robotics, and human-computer interaction.

**Possible Negative Societal Impacts:** 1) Privacy Concerns: The improved accuracy and efficiency of text recognition enabled by models like UNIT may raise concerns regarding privacy and data security. Increased capability to extract and analyze text from images could potentially infringe upon individuals' privacy rights if not properly regulated or safeguarded. 2) Job Displacement: The automation of document analysis tasks facilitated by UNIT and similar models could lead to job displacement in industries that rely heavily on manual data entry and document processing. While this could result in increased efficiency and cost savings for businesses, it may also contribute to unemployment and economic inequality.

