# OpenReview forum: "UNIT: Unifying Image and Text Recognition in One Vision Encoder"
_NeurIPS.cc/2024/Conference — NeurIPS 2024 poster_

### Official Review · Reviewer_rNyF · 2024-07-07

**Soundness:** 3
**Presentation:** 2
**Contribution:** 2
**Rating:** 5
**Confidence:** 4

**Summary:**

In this paper, the authors propose a novel training framework aiming at unify image and text recognition within a single model. The framework contains both visual and language decoders. In addition, the authors adopt a two-stage training, where the pre-training stage uses intra-scale data while the finetuning stage uses inter-scale samples. The experimental results demonstrate the effectiveness of the framework.

**Strengths:**

(1)	The authors aim to unify image and text recognition tasks in one module, which is good for the community.

(2)	The general pipeline of the paper is clear.

(3)	The performances improvement in the experiments seems significant.

**Weaknesses:**

(1)	Although the authors claim that they unify image and text recognition into one module, I find there are two different decoders. It seems the authors just combine two decoders in the framework, one for image recognition and another for text recognition. Hence, I think this unification is not very elegant.

(2)	The general pipeline is clear, but some details are not given. For example, the data preparation of both intra-scale and inter-scale training is not clear. In addition, it is not clear why the random feature sampling is effective. Because it will discard tokens, if those discarded tokens contain important information, the recognition will be definitely failed. By the way, I am not sure how many tokes will be discarded commonly.

(3)	In the paper description, the authors claim that the scale robustness is an important contribution (line 129), but I can not find sufficient experiments about this. In the experiments, I find the input scales are fixed, i.e., documents with 896\*896, and images with 224\*224 (I guess). Why not show more results of flexible inputs about the image and documents? Based on my understanding, the training data is about 896\*896 and 224\*224, so the test data will follow this resolution. If my understanding is correct, I suppose it can not be claimed about the scale robustness.

(4)	I can not understand the test stage. Does it mean the visual decoder is used for image recognition and the language decoder for text recognition? if it is correct, we need to specify the task for each image during the test stage, which will limit the usage. For example, if an image will be used for both image recognition and text recognition, how to deal with? In addition, this design is not flexible for users.

**Questions:**

(1)	In line 162, the authors describe that “low-resolution images, the position embedding are then randomly cropped”. I wonder why low-resolution images will need cropped instead of high-resolution images?

**Limitations:**

Not addressed yet.

---

> ### Author Rebuttal · Authors · 2024-08-07
>
> Thank you for recognizing the contribution of our model to the community. We will make it publicly available in the coming days. We understand that your main concerns and rating stem from some technical details and test settings. Below, we will address these issues one by one. We sincerely hope you can appreciate the significance of our work.
>
> ## **Q1. Unification of image and text recognition**
>
> As claimed in our title, we unify image and text recognition into **one single encoder** (not just "one module" as mentioned in your comment). This unification lies in the shared encoder, not the decoders. The two lightweight decoders primarily serve as auxiliary modules for training and are discarded during actual application.
>
> In our experiment, to evaluate image and text recognition abilities independently, we retain the vision decoder for zero-shot classification and the language decoder for OCR. We recommend users pair our encoder model with a large-scale decoder model (e.g., Vicuna-7B) to enhance both natural image and document-related tasks.
>
>
>
> ## **Q2. Some more details**
>
>
> **1. Data preparation**: We are unsure which specific part you find unclear. We have detailed the data sources, amounts, resolutions, and the ratio of natural images to documents in lines 207-212 and 233-241. I guess your confusion is about generating synthetic datasets of PDF documents, they are rendered with Latex/HTML codes from arXiv papers and websites, as in Nougat. If you have any other questions, please let us know.
>
> **2. Effectiveness of random feature sampling**:
>
> - **Ablation study**: In Table 4 of our paper, the results of "w/o feat. sampling" under "Exchanged Resolution", indicate a significant performance drop (31.66% vs. 80.49%) when the sampling strategy is disabled.
>
> - **Theoretical explaination**: Images have significant spatial redundancy; a missing patch can be recovered from neighboring patches through high-level understanding of parts, objects, and scenes. MAE[1] has demonstrated that even when 75% of image patch features are randomly dropped, the original image can still be reconstructed using a lightweight decoder.
>
> - **Advantages**: Randomly dropping vectors acts as a regularization technique, reducing noise in gradient calculations. This helps prevent overfitting, improves generalization, and leads to more stable and faster convergence.
>
>   [1] Masked Autoencoders Are Scalable Vision Learners, CVPR2022.
>
>
> ## **Q3. Scale Robustness**
>
> **1. Scale robustness**: This refers to the model's ability to maintain consistent perceptual capabilities across different input resolutions. Our model achieves scale robustness via inter-scale finetuning. The ablation study is presented in lines 312-316 and the 1st row in Table 4, which demonstrates a significant improvement in "Exchanged Resolution" (0.40% vs. 80.49%).
>
> **2. Input scales**: Our model can be trained with any resolution below the maximum limit. We chose two primary resolutions for training to ensure consistent tensor sizes, simplifying batch processing and optimizing resource use. UNIT is trained on these two resolutions and tested on different ones. The Table below shows zero-shot classification and OCR results at various resolutions, demonstrating that our model generalizes well and maintains consistent performance.
>
> |  Resolution | 224 | 336 | 448 | 672 | 896 |
> |  ----  | ----  | ---- | ----  | ----  | ----  |
> | Zero-shot cls. | 78.8 | 81.3 | 81.7 | 81.3 | 80.5 |
> | OCR F1 | 92.5 | 93.7 | 93.2 | 93.9 | 95.8 |
>
>   We also list some experiments in our paper that use different input size:
>
>   - In Table 4, we not only evaluate our model on zero-shot classification and OCR on their "Commonly used resolution" (image 224 and document 896), but also show the results on "Exchanged resolution" (image 896 document 224).
>   - As in line 278, the input resolution is set to 512x512 for semantic segmentation.
>
>
>
> **3. More experiments**: We also evaluated ViTs commonly used in MLLMs on zero-shot classification. These ViTs, pretrained on fixed resolutions, perform poorly as the difference between input size and pretraining resolution increases. In contrast, our model maintains consistent perceptual ability across different resolutions, with no sharp drop in performance.
>
>   |  Method  | 224 | 336 | 448 | 672 | 896 |
>   |  ----  | ----  | ---- | ----  | ----  | ----  |
>   | CLIP-L | 72.9 | 71.3 | 63.3 | 59.7 | 51.0 |
>   | OpenCLIP-H | 80.5 | 80.2 | 78.5 | 72.7 | 65.7 |
>   | EVACLIP-G  | 78.5 | 78.2 |  76.8 | 72.9 | 68.0 |
>   | UNIT | 78.8 | 81.3 | 81.7 | 81.3 | 80.5 |
>
>
>
> ## **Q4. Test setting**
>
> The two lightweight decoders primarily serve as auxiliary modules for training and are discarded during actual application. In our experiments, they are retained to independently compare image and text recognition abilities against state-of-the-art methods on zero-shot classification and OCR tasks.
>
> UNIT serves as a fundamental vision backbone, providing general visual embeddings for both natural images and documents. We recommend pairing our encoder with a large-scale decoder capable of handling both image and text information. For example, integrating the encoder with a LLM to form a MLLM supports a wide range of applications, including recognizing and understanding standalone images or documents, as well as scenarios where images and text appear together.
>
>
>
> ## **Q5. Position embedding**
>
> The position embeddings are set based on the model's maximum resolution. For example, if initialized to 64x64xd, where d is the hidden size, this corresponds to a maximum resolution of 896x896 with a patch size of 14. For an image of 896x896, the required position embeddings are 64x64xd, so cropping is unnecessary. However, for an image of 224x224, we need to randomly crop 16x16xd from the position embeddings.

---

> > ### Author Response · Authors · 2024-08-11
> >
> > Dear reviewer,
> >
> > We have provided detailed clarifications on all your questions point to point. If you have any additional questions or suggestions, we would be very happy to discuss them with you. We sincerely hope you can appreciate the significance of our work and consider updating your rating. Thank you!

---

> > ### Comment · Reviewer_rNyF · 2024-08-13
> >
> > I am sorry that it is a little late for the discussion. Thanks authors for the responses. These responses really address some of my concerns, like the usage of encoder and decoder. However, I still can not understand some of your answers, like Q5, why crop 16*16*d for 224*224, but never mind, I can read the original paper CPE for more details.
> >
> > Another question is your answer in Q2 about the random feature sampling. Although you give an example of the MAE paper, it is about the natural image, which is very different with the document image. I do not believe that dropping 75% of patches can be predicted for the document images, because the text patterns are much more complicated than the general objects, like cat, dog.

---

> ### Author Response · Authors · 2024-08-13
>
> Dear reviewer,
>
> Thank you for your thoughtful comments and for engaging in a detailed discussion.
>
> **Q2. About random feature sampling**
>
> We would like to clarify that the random feature sampling approach is indeed applied **only to natural images**, not document images. As mentioned in lines 242-244 of our paper, this technique is specifically designed to enhance training stability and efficiency when computing the token-wise feature reconstruction loss for high-resolution natural images. The purpose of this feature reconstruction loss is to maintain the original image recognition capability, and it is solely utilized for natural images. Therefore, the concern you have about random sampling causing a loss of text information will not occur.
>
> **Q5. About position embedding**
>
> Our position embedding is 64x64xd, but for input sizes smaller than 896x896, such as 224x224, a 16x16xd position embedding is required. There are generally three possible solutions: cropping, interpolation, and training another 16x16xd position embedding. We chose cropping for the following reasons:
>
> 1. Cropping is a more straightforward approach compared to other methods, as it avoids additional computational costs and memory usage.
> 2. As mentioned in lines 238-239 of our paper, the low-resolution document dataset consists of images cropped from high-resolution documents. Therefore, we use cropped position embeddings for low-resolution inputs to preserve local structural information, ensuring consistent spatial feature representation across resolutions. For natural images, both interpolation and cropping are viable, and we selected cropping for its simplicity.
>
> We hope this explanation addresses your concerns. If our response has clarified the issues and alleviated your concerns, we would sincerely appreciate it if you could kindly consider updating your score.

---

> > ### Comment · Reviewer_rNyF · 2024-08-14
> >
> > thanks for your quick response. Then It make sense for random feature sampling. Since the Table 4 shows the text recognition ability, it very easy to make readers think this is also for document images. I suggest the authors can highlight the sampling for natural images. I plan to increase the rating, but I strongly suggest the authors to make the paper more clear and highlight the contributions.

---

> > > ### Author Response · Authors · 2024-08-14
> > >
> > > Dear reviewer,
> > >
> > > Thank you for your prompt and thoughtful feedback. We appreciate your understanding regarding the random feature sampling. Following your suggestion, we will make sure to highlight that the sampling is specifically for natural images.
> > >
> > > We are also committed to revising the paper to ensure that our contributions are clearly presented. Your input has been invaluable, and we're grateful for your increased score.
> > >
> > > Thank you once again for your time and support!

---

### Official Review · Reviewer_W6eH · 2024-07-09

**Soundness:** 3
**Presentation:** 3
**Contribution:** 3
**Rating:** 5
**Confidence:** 4

**Summary:**

This paper presents a new visual backbone training framework named UNIT, which integrate image and text recognition simultaneously. The UNIT framework leverages text recognition, visual feature construction and image captioning as the pre-training tasks. The training pipeline includes two stage: intra-scale pre-training and inter-scale pre-training with different solution for different types of images. Extensive experiments demonstrates UNIT significantly outperforms existing methods on document-related tasks while maintaining performance on natural images without additional inference cost.

**Strengths:**

1. UNIT enables the visual backbone to recognize both text and natural images without the need for additional visual experts.

2. The proposed model retains the original Vision Transformer architecture, making it cost-effective in terms of inference and deployment for various types of images.

3. The two pre-training stages are well-suited for document image pre-training, which typically requires high resolution.

4. The experiments demonstrate that UNIT achieves state-of-the-art performance on various document-related tasks while maintaining strong performance on natural image tasks.

**Weaknesses:**

1. The experiments are insufficient for understanding how the training pipeline works. The paper only leverages resolutions of 1x and 4x and does not show results for training with different resolutions.

2. Document images tend to be non-square. During the pre-training stage, using square images might hinder learning for documents with extreme aspect ratios, such as those found in InfoQA [1]. This issue is not discussed in the paper.

3. The model/data/resolution scaling behaviors of the proposed UNIT is not discussed which is important for pre-training framework.

[1] Mathew, Minesh, et al. "Infographicvqa." Proceedings of the IEEE/CVF Winter Conference on Applications of Computer Vision. 2022.

**Questions:**

1. In Table 3, we observe that even when using image captioning during pre-training, the zero-shot classification performance on natural images remains poor without applying feature reconstruction as the pre-training task. I am curious as to why this approach fails, given that Cappa [1] also utilizes image captioning for pre-training and scales well in natural image classification. Could you provide a detailed discussion on this discrepancy?

2. For the LLaVA experiments, it seems that two different resolutions are applied for different types of images. Why not considering using both 224 or 896 resolution? Is there any negative signals for using large image resolutions for natural image?

3. For the language decoder, using Q-former would be not necessary and not trivial. It seems that using naive MLP layers can also work.

[1] Tschannen, Michael, et al. "Image captioners are scalable vision learners too." Advances in Neural Information Processing Systems 36 (2024).

**Limitations:**

The authors have addressed the limitation in the manuscripts.

---

> ### Author Rebuttal · Authors · 2024-08-07
>
> Thank you for the constructive comments. We understand your main concerns are related to resolution settings and scaling behaviors. We will address these issues one by one and hope you can recognize the significance of our work.
>
> ## **Q1. Training with different resolutions**
>
> Our model can be trained with any resolution below the maximum limit. We chose two primary resolutions for training to ensure consistent tensor sizes, simplifying batch processing and optimizing resource use. UNIT is trained on these two resolutions and tested on different ones. The Table below shows zero-shot classification and OCR results at various resolutions, demonstrating that our model generalizes well and maintains consistent performance.
>
> |  Resolution | 224 | 336 | 448 | 672 | 896 |
> |  ----  | ----  | ---- | ----  | ----  | ----  |
> | Zero-shot cls. | 78.8 | 81.3 | 81.7 | 81.3 | 80.5 |
> | OCR F1 | 92.5 | 93.7 | 93.2 | 93.9 | 95.8 |
>
>
>
> ## **Q2. Different document input size**
>
> To ensure efficient data loading and processing during training, we preprocessed the document images by padding them to square shapes. During inference, we maintain the original aspect ratio of the input images.
>
>
> ## **Q3. Scaling behaviors**
>
>
> **1. Model scaling:** In our paper, we primarily present the UNIT model with 0.6B parameters. To scale up the model, we added 18 more transformer layers, creating a new version with 1B parameters. After training this UNIT-1B model using the same settings and data, we observed further improvements in almost all the benchmarks.
>
> | Model | Zero-shot Cls. | OCR Prec. | OCR Rec. | OCR F1 |
> |  ----  | ----  | ---- | ----  | ---- |
> | UNIT-0.6B | 78.8 | 96.7 | 94.0 | 95.3 |
> | UNIT-1B | 79.2 | 98.7 | 95.6 | 98.9 |
>
> | Model | VQAv2 | GQA | OKVQA | DocQA | ChartQA | InfoVQA |
> |  ---- | ---- | ----  | ---- | ---- | ----  | ----  |
> | UNIT-0.6B | 72.3 | 60.4 | 55.5 | 43.0 | 46.0 | 28.7 |
> | UNIT-1B | 76.6 | 62.9 | 56.7 | 69.6 | 62.2 | 37.9 |
>
>
> **2. Data scaling:**
>
> - More Natural Image Data: We added 5M LAION-COCO data to $\mathcal D^I_{\times 1}$. Increasing the description data of natural images did not lead to significant performance gains, as the original CC3M dataset already covers most common visual concepts in downstream VQA tasks.
>
> - More Document Data: We added 1M document data (from Nougat) to $\mathcal D^T_{\times 4}$, resulting in a slight improvement in document-oriented QA tasks, as the new data provided more diverse samples for different fonts and sizes.
>
> - Finetuning with SFT Data: Beyond freezing the encoder model as done in Llava experiments, we further finetuned the encoder model with the LLM during SFT. This led to significant performance improvements, especially in document-oriented QA tasks, indicating the necessity of high-quality instruction data.
>
>     | Setting | VQAv2 | GQA | OKVQA | DocQA | ChartQA | InfoVQA |
>     | ----  | ---- | ----  | ---- | ---- | ----  | ----  |
>     | UNIT(ours) | 72.3 | 60.4 | 55.5 | 43.0 | 46.0 | 28.7 |
>     | More natural image data | 72.5 | 60.5 | 54.9 | 42.4 | 46.3  | 28.3 |
>     | More document data | 71.9 | 59.7 | 54.3 | 44.5 | 47.7 | 29.8 |
>     | Finetune with SFT data | 76.7 | 63.4 | 56.9 | 67.6 | 63.4 | 36.2 |
>
>
> **3. Resolution scaling:** Scaling up the maximum resolution increases computational costs for training the encoder with higher-resolution images. To address this, we follow existing document MLLMs (mPLUG-DocOwl, Qwen-VL, and Monkey) by using a grid slicing strategy, dividing each image into grids. The encoder generates visual tokens for each grid, which are then concatenated to form the final representation. Despite using LLMs with similar parameters, our method achieves superior performance, as shown in the table:
>
> |  Method  | Vision Encoder  | DocVQA | ChartVQA | InfoVQA |
> |  ----  | ----  | ---- | ----  | ----  |
> | mPLUG-DocOwl | CLIP-L 0.3B  | 62.2 | 57.4 | 38.2 |
> | Qwen-VL | ViT-BigG 1.9B | 65.1 | 65.7 | 35.4 |
> | Monkey  | ViT-BigG 1.9B |  66.5 | 65.1 | 36.1 |
> | **Ours** | UNIT 0.6B |  **68.3** | **66.2** | **39.1** |
>
> ## **Q4. Zero-shot cls results in Table 3**
>
> This result is reasonable because zero-shot classification evaluates the alignment between the vision and text encoders. When the feature reconstruction loss $\lambda=0$, the vision encoder cannot ensure proper alignment with the text encoder of OpenCLIP-H, leading to the failure of zero-shot classification. CapPa does not use the CLIP models' text encoder for zero-shot classification. Instead, it uses LiT[1], an efficient method to equip any pretrained vision backbone with zero-shot classification, to learn a text embedding that matches the embedding of their pretrained vision encoders.
>
> - [1] LiT: Zero-Shot Transfer with Locked-image text Tuning.
>
> ## **Q5. About LLaVA experiments**
>
> The resolution is determined by task requirements, not image type. Low resolution is used for global understanding (e.g., VQA), while high resolution is used for fine-detail perception (e.g., DocQA). The token number is more important and effective than the resolution. Using a 896 resolution for natural images does not improve performance, but increasing the token number to 576 can enhance it.
>
> |  Resolution | Token num | VQAv2 | GQA | OKVQA |
> |  ----  | ----  | ---- | ----  | ----  |
> | 224 | 256 | 72.3 | 60.4 | 55.5 |
> | 896  | 256 | 71.0 | 60.0 |  54.3 |
> | 896  | 576 | 74.0 | 62.1 |  55.8 |
>
>
> ## **Q6. Replace Q-former**
>
> Q-former reduces the number of visual tokens for images. An image of 896x896 generates 4096 tokens, which is too computationally intensive for a lightweight language decoder and slows convergence. In our training framework, Q-former should be replaced with modules like pooling layers that reduce token count. However, when integrated with a large-scale LLM like Llama-7B, which supports long input sequences, Q-former can be replaced with MLP layers.

---

> > ### Author Response · Authors · 2024-08-11
> >
> > Dear reviewer,
> >
> > Thank you for your valuable comments. As the deadline is approaching, we kindly inquire whether our discussions have addressed your concerns. If you have any further questions, we would be happy to continue our conversation. If our response has resolved your concerns, we would greatly appreciate it if you could consider updating your score and providing feedback. Thank you again.

---

> ### Comment · Reviewer_W6eH · 2024-08-12
>
> After reading the authors' feedback and other reviewers' opinions, I would like to thank the authors for their rebuttal.
>
> The rebuttal addresses most of my concerns, and thus, I raise my score. I would like to note that even though the authors addressed my concerns, I believe that the paper needs some sort of revision, as it appears that many basic concerns were needed to be clarified in the rebuttal. I strongly encourage the authors to fix the main comments mentioned by the reviewers.

---

> > ### Author Response · Authors · 2024-08-12
> >
> > Dear reviewer,
> >
> > Thank you very much for reconsidering our paper and for raising your score. We appreciate your recognition of our efforts to address your concerns.
> >
> > Your feedback has been instrumental in enhancing the quality of our submission, and we are grateful for the time and thoughtfulness you have dedicated to your review. We have carefully considered your suggestions and are committed to revising the paper accordingly. We believe these revisions will further strengthen our submission.

---

### Official Review · Reviewer_qQPk · 2024-07-11

**Soundness:** 2
**Presentation:** 3
**Contribution:** 3
**Rating:** 6
**Confidence:** 4

**Summary:**

The paper proposes to enable vision encoders to support general image recognition and text recognition at the same time, in which a lightweight language decoder is proposed for predicting text outputs and a lightweight vision decoder is proposed to prevent catastrophic forgetting of the original image encoding capabilities. Besides the intra-scale pertaining, an inter-scale fine-tuning stage is utilized to improve the robustness of both image recognition and text recognition. Experiments on multiple datasets demonstrate the effectiveness of the proposed method and its potential to process interleaved text and image information in real-world scenarios.

**Strengths:**

1) The paper proposes to integrate strong text recognition and image recognition into one encoder, which provides good potential for complex document analysis and general vision recognition.
2) The proposed techniques including inter-scale training and feature sampling help the model to deal with diversities for image recognition and text recognition.
3) Experiments on several benchmarks demonstrate the superiority over general encoders (CLIP, DINO) and specialized document models (vary, nougat).

**Weaknesses:**

1) The ability to keep ability from multiple sources is mainly achieved by distillation from a teacher model, which follows existing works like RADIO and is not quite new.
2) As stated in L300-303, the teachers used for other models are different from UNITS, which may lead to unfair comparison.
3) How does the model effectively processing high-resolution document images with the transformer vision encoder?
4) Missing comparison with existing document MLLM, e.g., Monkey.
[1] Monkey: Image resolution and text label are important things for large multi-modal models, CVPR 2024.
5) Missing reference for Figure 2. Incomplete information in the references part, e.g.,  reference [6] A. Brock, S. De, S. L. Smith, and K. Simonyan. High-performance large-scale image recognition without normalization, 2021.

**Questions:**

Please refer to the weakness part.

**Limitations:**

Yes. Positive societal impacts and possible negative societal impacts are well described in the A.3 section.

---

> ### Author Rebuttal · Authors · 2024-08-07
>
> Thank you for your comprehensive and encouraging review. We are pleased to hear that you appreciate the technical contributions and the significantly improved performance of our proposed visual encoder, UNIT, along with the intra/inter-scale training paradigm. In the following, we will respond to your concerns and questions:
>
> ## **Q1. Compare with RADIO**
>
> The main contribution of this paper is the successful integration of text recognition capabilities into a vision encoder that was pretrained solely on natural images, while still preserving its original image recognition abilities. In our method, the text recognition capability is integrated via a language decoding process, and the image recognition ability is preserved through model distillation. This approach differs from pure model distillation methods like RADIO in two key aspects:
>
> 1. **Training tasks:** Our method employs a multi-task training framework that jointly optimizes distillation, image captioning, and OCR tasks. This necessitates careful model design and implementation to balance these different objectives, making the approach more challenging than optimizing solely with distillation tasks, as done in RADIO.
>
> 2. **Capability upper bound:** In pure distillation methods, the teacher models represent the upper limit of the student's capabilities (e.g., RADIO's performance on zero-shot classification and segmentation is within 3% of that of their teacher models), which may limit the student's learning potential. In contrast, our method learns text recognition from millions of data samples, allowing it to surpass state-of-the-art text recognition models by significant margins, ranging from 2% to 42%.
>
>
>
> ## **Q2. Experimental settings about teacher models**
>
> The compared models use DINOv2 and OpenCLIP-H as their teachers, while we use RADIO-H, which is also distilled from DINOv2 and OpenCLIP-H. Therefore, both the compared models and our model **share the same foundational teacher knowledge** derived from DINOv2 and OpenCLIP-H. We believe this makes the comparison reasonable.
>
>
> ## **Q3. High-resolution document inputs**
>
> UNIT supports input images with a maximum size of 896x896. For images with resolutions higher than this, we employ a grid slicing strategy, where each image is divided into several grids according to their aspect ratio. For example, when the maximum grid number is set to 4, the grids could be {2x2, 1x{2,3,4}, {2,3,4}x1}, each of size 896x896. The encoder generates visual tokens for each grid, which are then concatenated as the final representation of the image.
>
> This slicing strategy fragments the original input, disrupting contextual continuity and spatial geometry across patches. However, training the encoder directly with high-resolution images incurs extremely high computational costs. To balance these factors, we should minimize the number of splits. Our method extends the supported resolution range compared to conventional ViTs, with a maximum resolution of 896x896.
>
>
> ## **Q4. Comparison with existing document MLLM**
>
> We create an MLLM using UNIT as the vision encoder. During training, we utilize the CC3M dataset along with 2 million document images for pretraining, and SFT data from LLAVA-665K, including the training sets of DocQA and ChartQA. The document images are divided into grids of size 448x448 to ensure consistency with Monkey[1]. We compare our method with existing document MLLMs, including Monkey[1], Qwen-VL[2], and mPLUG-DocOwl[3], all of which use 7B LLM models. As shown in the table below, our method achieves superior performance on document-oriented VQA tasks.
>
> |  Method  | Vision Encoder  | DocVQA | ChartVQA | InfoVQA |
> |  ----  | ----  | ---- | ----  | ----  |
> | mPLUG-DocOwl | CLIP-L 0.3B | 62.2 | 57.4 | 38.2 |
> | Qwen-VL | ViT-BigG 1.9B | 65.1 | 65.7 | 35.4 |
> | Monkey  | ViT-BigG 1.9B |  66.5 | 65.1 | 36.1 |
> | **Ours** | UNIT 0.6B |  **68.3** | **66.2** | **39.1** |
>
> The main difference between these MLLMs and our method is that their vision encoders are pretrained solely on natural images and are trained with LLMs directly on document QA tasks by adding LoRA [1] or through direct finetuning [2,3]. In contrast, UNIT, as a vision encoder, is pretrained on both natural images and documents with fundamental recognition tasks, i.e., image captioning and OCR. The advantage lies in two aspects: First, pretraining on fundamental recognition tasks makes UNIT more versatile and better suited for downstream document tasks. Second, UNIT's early integration of text recognition capabilities provides a robust initialization for document MLLM training, resulting in more stable training and faster convergence.
>
> To address your concern, we will add these discussions in the revised manuscript.
>
> - [1] Monkey: Image resolution and text label are important things for large multi-modal models, CVPR 2024.
> - [2] Qwen-VL: A Versatile Vision-Language Model for Understanding, Localization, Text Reading, and Beyond.
> - [3] mPLUG-DocOwl: Modularized Multimodal Large
> Language Model for Document Understanding
>
>
>
> ## **Q5. Reference and citation details**
>
> Thank you for your valuable suggestions. We will address the issues as follows:
>
> 1. The reference for Figure 2 will be added in the appropriate sections, specifically in lines 204 and 232.
> 2. We acknowledge the incomplete information in the references. The citation for the paper you mentioned, "[6] A. Brock, S. De, S. L. Smith, and K. Simonyan. High-performance large-scale image recognition without normalization, 2021," will be updated with the complete details.
>
> We appreciate your attention to detail and will ensure these corrections are made promptly.

---

> > ### Comment · Reviewer_qQPk · 2024-08-12
> > **Official Comment by Reviewer qQPk**
> >
> > Thank the authors for their response.
> >
> > Regarding W1, in fact, the proposed method integrates both distillation in works like RADIO and supervised learning widely used in OCR methods, e.g., vary, which is not originally designed.
> >
> > Regarding W2-W5, I appreciate the response and believe the details serve as a good supplement to the original manuscript.
> >
> > I'd like to maintain the rating in the current stage.

---

> ### Author Response · Authors · 2024-08-12
>
> Dear Reviewer,
>
> Thank you for your thorough review and constructive feedback throughout the process. We appreciate your recognition of our efforts to address your concerns.
>
> We have carefully considered your suggestions and are committed to revising the paper as you recommended. These revisions will further strengthen our submission, and we are grateful for your insights that have guided us in this direction.
>
> Thank you again for your support!

---

### Author Rebuttal · Authors · 2024-08-07

We thank all the reviewers for their time, insightful suggestions, and valuable comments. We respond to each reviewer's comments in details. These issues will be addressed in the revised manuscript, and we believe this makes our paper much stronger.

We promise to release all the pre-trained models and their codes to promote the development of vision foundation models and MLLMs. We believe this will significantly contribute to the community by providing valuable resources for research and innovation.

---

### Decision · Program_Chairs · 2024-09-25

**Decision:**

Accept (poster)

**Comment:**

The manuscript received originally mixed reviews. Nevertheless, the authors response clarified many aspects raised by the reviewers, leading to positive final reviews from all reviewers.

A lot of detailed information was presented by the authors during the rebutal. Although I consider the paper worth presenting in the conference, it is important for these clarifications provided during the reviewing process to be properly integrated in the revised version of this manuscript.